# Epidermal Barrier Function and Skin Homeostasis in Atopic Dermatitis: The Impact of Age

**DOI:** 10.3390/life12010132

**Published:** 2022-01-17

**Authors:** Trinidad Montero-Vilchez, Carlos Cuenca-Barrales, Juan-Angel Rodriguez-Pozo, Pablo Diaz-Calvillo, Jesús Tercedor-Sanchez, Antonio Martinez-Lopez, Alejandro Molina-Leyva, Salvador Arias-Santiago

**Affiliations:** 1Dermatology Department, Hospital Universitario Virgen de las Nieves, Avenida de Madrid, 15, 18012 Granada, Spain; tmonterov@correo.ugr.es (T.M.-V.); carloscuenca1991@gmail.com (C.C.-B.); juanangelrpg@gmail.com (J.-A.R.-P.); pdc.muro@gmail.com (P.D.-C.); jesustercedor@gmail.com (J.T.-S.); antoniomartinezlopez@aol.com (A.M.-L.); salvadorarias@ugr.es (S.A.-S.); 2Instituto de Investigación Biosanitaria Granada, 18012 Granada, Spain; 3Dermatology Department, Faculty of Medicine, University of Granada, 18011 Granada, Spain

**Keywords:** aging, atopic dermatitis, skin barrier, stratum corneum

## Abstract

Skin is damaged in atopic dermatitis (AD) patients. Age is also believed to have a negative effect on epidermal barrier function. The aim of this study was to investigate skin barrier function changes with age in AD patients. A cross-sectional study was conducted including 162 participants, 81 AD patients and 81 healthy volunteers. Skin barrier function parameters, such as transepidermal water loss (TEWL), erythema, temperature, stratum corneum hydration (SCH), pH, and elasticity, were evaluated. Healthy volunteers were evaluated on the volar forearm. AD patients were measured on two regions: on an eczematous lesion on the volar forearm and on a non-involved area 5 cm from the affected area. TEWL was lower on healthy skin than uninvolved AD skin (9.98 vs. 25.51 g·m^−2^·h^−1^, *p* < 0.001) and AD eczematous lesions (9.98 vs. 28.38 g·m^−2^·h^−1^, *p* < 0.001). SCH was lower on AD eczematous lesions than uninvolved AD skin (24.23 vs. 39.36 AU, *p* < 0.001) and healthy skin (24.23 vs. 44.36 AU, *p* < 0.001). Elasticity was lower on AD eczematous lesions than uninvolved AD skin (0.69 vs. 0.74, *p* = 0.038) and healthy skin (0.69 vs. 0.77, *p* = 0.014). A negative correlation was found between age and elasticity in all the population (r = −0.383, *p* < 0.001). This correlation was stronger in AD patients (r = −0.494, *p* < 0.001) than in controls (r = −0.266, *p* = 0.092). After conducting a linear regression model in AD patients adjusted by age, sex, and SCORing Atopic Dermatitis (SCORAD), it was found that elasticity was impaired by an increasing age (β = −0.004, *p* < 0.001) and a higher SCORAD (β = −0.003, *p* < 0.001). The skin barrier function is impaired by age and AD, reflected mainly in poor elasticity values in older AD patients.

## 1. Introduction

The skin is the human body’s largest organ. It accomplishes multiple defensive and regulatory functions as it protects the body against external stressors and maintains cutaneous homeostasis [1]. The skin barrier function resides mainly in the stratum corneum of the epidermis [2]. Transepidermal water loss (TEWL) evaluates the diffusion of condensed water through the stratum corneum and is a key characteristic of the skin barrier [3]. Greater TEWL is often associated with skin barrier impairment and has been observed in several skin diseases [4,5]. Stratum corneum hydration (SCH), the water content of the stratum corneum, is another important parameter, and a lower value is frequently associated with skin barrier dysfunction [6]. The skin surface pH is also considered in the assessment of epidermal functions, as high pH values are related to loss of antimicrobial activity [7]. Erythema is also useful in assessing the integrity of the epidermal barrier [8]. Elasticity is another important feature related to skin biomechanical properties [9].

Skin aging is an intrinsic and extrinsic process. Intrinsic, or chronologic aging, is an inevitable, genetically determined process. Histologically, the epidermis gets thinner, and the dermal-epidermal junction flattens, increasing skin fragility and decreasing nutrient transfer. Epidermal cell turnover decreases, slowing wound healing, and the dermis becomes atrophic with reduced numbers of fibroblasts and subdermal adipose tissue [10]. Moreover, the number and diameter of collagen fiber bundles decrease and the ratio of type III collagen to type I collagen increases [11]. Clinically, intrinsic aging is characterized by laxity and some exaggerated expression lines [12]. Extrinsic aging is associated with external factors, with sun exposure being the most deleterious. Histologically, photoaged skin is characterized by elastosis, an accumulation of elastin material below the dermal-epidermal junction, epidermal atrophy, and fragmentation of collagen and elastic fibers. Clinically, it is reflected as dryness, wrinkles, irregular pigmentation, loss of elasticity, telangiectasias, and purpura [13].

Atopic dermatitis (AD) is a chronic cutaneous inflammatory disease caused by environmental and genetic factors. It is characterized by recurrent, eczematous lesions associated with pruritus [14]. Epidermal barrier dysfunction, immune dysregulation, and gut dysbiosis may play roles in this disease [15]. AD affects up to 20% of children and 10% of adults, with higher prevalence in industrialized countries [16]. Nevertheless, the incidence in adult patients is increasing due to an aging society and the accumulation of environmental stressors and their cumulative impact on epidermal barrier function [17]. There is scarce evidence regarding the differences between AD in children and adults [18]. Children compared with adults with AD showed decreased filaggrin expression and activation of T helper cells (Th)2, Th22, and Th1, and a higher induction of Th17-related cytokines, antimicrobials, Th9, interleukin (IL)-33, and innate markers [19]. Moreover, it was observed that IL-31 and IL-33 levels were higher in children than adults with AD, whereas thymic stromal lymphopoietin (TSLP) and immunoglobulin (Ig) E levels were similar in children and adults [20]. Nevertheless, we found no studies that evaluated the differences in epidermal barrier function between adults with AD younger and older than 30 years. Assessing skin homeostasis and epidermal barrier function in lesioned and non-lesioned skin would provide a better understanding of the complex pathogenesis of this disease [21], as well as would provide tools to assess disease severity objectively [5].

Skin aging shares some immunological findings with AD. Both Th1 and Th17 are increased with age and in AD patients. Nevertheless, Th2/Th22 and matrix metalloproteinase 12 (MMP-12) increase within normal aging while their levels are inversely correlated with age in the skin of older AD patients [22,23]. The terminal differentiation markers expression, such as filaggrin or loricrin, significantly increase with age in AD, while they decrease in endogenously aged skin [17]. Ki16 and Ki67, epidermal hyperplasia markers, are increased in atopic skin but they diminish in AD with age [24]. To date, the differences between endogenous skin aging and aging AD skin are not completely understood.

The aim of this study is to investigate skin barrier function changes with age in AD patients, assessed by objective parameters including TEWL, SCH, erythema, temperature, pH and elasticity.

## 2. Materials and Methods

A cross-sectional study was conducted. Participants were recruited from October 2020 to February 2021 in the Dermatology Service of the Hospital Universitario Virgen de las Nieves, Granada, Spain.

Patients with an established clinical diagnosis of mild to severe AD [14] were included in the study. The diagnosis of AD was made by a dermatologist following Hanifin y Rajka criteria [14,25]. Healthy controls, volunteers who attended our Dermatology Department for common conditions, such as seborrheic keratoses or melanocytic nevi, and did not have previous family or personal history of any inflammatory skin disease were also included in the study. The exclusion criteria were having a clinical infection on the measured area, history of cancer or not signing the informed consent form.

Sociodemographic and clinical data were gathered by clinical interview and physical examination. Sex, age, smoking/alcohol habits, and emollient use were collected. The participants were classified according to their age: <30 years or ≥30 years, as the turning points of skin barrier function appears in an individual’s thirties [26]. AD severity was assessed by the SCORing Atopic Dermatitis (SCORAD), the Eczema Area Severity Index (EASI), and body surface area (BSA).

Homeostatic parameters in relation to epidermal barrier function were evaluated on the volar forearm. AD patients were measured on an eczematous area on the volar forearm and on a non-involved area 5 cm from the affected area. Healthy volunteers were also measured on the volar forearm. TEWL (in g·m^−2^·h^−1^, using Tewameter ^®^ TM300, Mirocaya, Bilbao, Spain), pH (using Skin-pH-Meter ^®^ PH905, Mirocaya, Bilbao, Spain), skin temperature (in °C, using Skin-Thermometer ST 500, Mirocaya, Bilbao, Spain), erythema index (in arbitrary units (AU), using Mexameter ^®^ MX 18, Mirocaya, Bilbao, Spain), SCH (in arbitrary units, using Corneometer ^®^ CM825, Mirocaya, Bilbao, Spain), and elasticity parameters (including R2 value, measured in %, using Cutometer ^®^ Dual MPA 580, Mirocaya, Bilbao, Spain) were measured by a Multi Probe Adapter (MPA, Courage + Khazaka electronic GmbH, Mirocaya, Bilbao, Spain). All these measurements were taken following the same order. The parameters were measured ten times and their average was used for analysis. All these measurements were taken in the same room at a mean room ambient air humidity of 45% (range, 40–50%) and temperature of 22 ± 1 °C. All participants underwent an adaptation period of at least 20 min before the measurements were taken. Topical or systemic treatments were not allowed three hours before the measurements were taken.

To perform the descriptive analysis, qualitative variables were expressed as absolute and relative frequency distributions and continuous variables as means ± standard deviations (SD). To compare continuous variables, the Student’s *t*-test for independent samples or Student’s *t*-test for paired samples were used, as appropriate. To test for possible correlations between continuous variables, the Pearson correlation coefficient was calculated. Linear regression models were used to evaluate factors associated with impaired cutaneous homeostasis. Statistical significance was defined by a two-tailed *p* < 0.05. SPSS version 24.0 (SPSS Inc., Chicago, IL, USA) was used for statistical analyses.

This study was approved by the ethics committee of Hospital Universitario Virgen de las Nieves (Epidermal Barrier Function and Skin Homeostasis project). The nature of the study was explained to all the participants, who agreed to participate and signed their informed consent form. All measurements were non-invasive, and the confidentiality of participant data was strictly preserved.

## 3. Results

This study included 162 participants, 81 patients with AD and 81 healthy volunteers with a mean age of 29.64 (16.71 SD) years, Figure 1. The sociodemographic characteristics of the sample are described in Table 1. AD patients had a mean EASI of 19.37 (8.59 SD) and a mean SCORAD of 40.98 (21.48 SD).

Skin homeostasis parameters between healthy skin, uninvolved AD skin and AD eczematous lesions were compared (Figure 2, Appendix A). TEWL was lower on healthy skin than uninvolved AD skin (9.98 vs. 25.51 g·m^−2^·h^−1^, *p* < 0.001) and AD eczematous lesions (9.98 vs. 28.38 g·m^−2^·h^−1^, *p* < 0.001), while no differences between uninvolved AD skin and AD eczematous lesion were found. SCH was lower on AD eczematous lesions than uninvolved AD skin (24.23 vs. 39.36 AU, *p* < 0.001) and healthy skin (24.23 vs. 44.36 AU, *p* < 0.001). Temperature was higher on AD eczematous lesions compared with uninvolved AD skin (32.07 vs. 31.28 °C, *p* < 0.001) and healthy skin (32.07 vs. 31.11 °C, *p* < 0.001). Erythema was higher on AD eczematous lesion than uninvolved AD skin (391.11 vs. 239.88 AU, *p* < 0.001) and healthy skin (391.11 vs. 218.93 AU, *p* < 0.001). pH was higher on AD eczematous lesions compared with healthy skin (6.15 vs. 5.92, *p* = 0.039). Elasticity was lower on AD eczematous lesions than uninvolved AD skin (0.69 vs. 0.74, *p* = 0.038) and healthy skin (0.69 vs. 0.77, *p* = 0.014).

Ninety-three participants were <30 years and 69 were ≥30 years of age. Sex, smoking/drinking habits, and emollients use was similar between the two age groups. Disease severity was compared between AD patients <30 and ≥30 years, Figure 3. AD patients ≥30 years had a tendentially higher EASI than AD patients <30 years (21.96 vs. 17.68, *p* = 0.090) while there were no differences regarding SCORAD (45.57 vs. 38.39 respectively, *p* = 0.166).

Regarding participants <30 years of age (Appendix A), TEWL was lower on healthy skin than uninvolved AD skin (9.99 vs. 25.69 g·m^−2^·h^−1^, *p* < 0.001) and AD eczematous lesions (9.99 vs. 28.32 g·m^−2^·h^−1^, *p* < 0.001) but it did not differ between AD eczematous lesions and uninvolved AD skin (Figure 4A). SCH was lower on AD eczematous lesions than uninvolved AD skin (22.47 vs. 38.22 AU, *p* < 0.001) and healthy individuals (22.47 vs. 48.13 AU, *p* < 0.001) and it was also lower on uninvolved AD skin than healthy individuals (38.22 vs. 48.13 AU, *p* = 0.030) (Figure 5A). Temperature was higher on AD eczematous lesions than uninvolved AD skin (31.94 vs. 31.05, *p* < 0.001) and healthy skin (31.94 vs. 31.07 °C, *p* = 0.010) but it did not differ between uninvolved AD skin and healthy skin (Figure 6A). Erythema was higher on AD eczematous lesions than uninvolved AD skin (388.80 vs. 226.75 AU, *p* < 0.001) and healthy skin (388.80 vs. 226.73 AU, *p* < 0.001), but it did not differ between uninvolved AD skin and healthy skin (Figure 7A). pH was lower on healthy volunteers than uninvolved AD skin (5.81 vs. 6.16, *p* = 0.029) and AD eczematous lesions (5.81 vs. 6.23, *p* = 0.008) but it did not differ between uninvolved AD skin and AD eczematous lesions (Figure 8A). Elasticity was lower on AD eczematous lesions than uninvolved AD skin (0.71 vs. 0.78, *p* = 0.003) and on healthy skin (0.71 vs. 0.79, *p* = 0.040) but it did not differ between uninvolved AD skin and healthy skin (Figure 9A).

Concerning adults ≥ 30 years of age (Appendix A), TEWL was lower on healthy skin than uninvolved AD skin (10.75 vs. 25.18 g·m^−2^·h^−1^, *p* = 0.001) and AD eczematous lesions (10.75 vs. 28.48 g·m^−2^·h^−1^, *p* < 0.001) but it did not differ between uninvolved AD skin and AD eczematous lesions (Figure 4B). SCH was lower on AD eczematous lesions than on uninvolved AD skin (27.63 vs. 41.56 AU, *p* = 0.001) and healthy skin (27.63 vs. 43.16 AU, *p* < 0.001) but it did not differ between uninvolved AD skin and healthy skin (Figure 5B). Temperature was higher on AD eczematous lesions than uninvolved AD skin (32.30 vs. 31.70 °C, *p* = 0.011) and healthy skin (32.30 vs. 31.17 °C, *p* < 0.001) and it was also higher on uninvolved AD skin than healthy skin (31.70 vs. 31.17 °C, *p* = 0.042) (Figure 6B). Erythema was higher on AD eczematous lesions than uninvolved AD skin (395.32 vs. 263.82 AU, *p* < 0.001) and healthy skin (395.32 vs. 214.08 AU, *p* < 0.001) and it was also higher on uninvolved AD skin than healthy skin (263.82 vs. 214.08 AU, *p* = 0.004) (Figure 7B). Elasticity was higher on healthy skin than uninvolved AD skin (0.74 vs. 0.65, *p* = 0.043) and AD eczematous lesions (0.74 vs. 0.65, *p* = 0.040) but it did not differ between AD eczematous lesions and uninvolved AD skin (Figure 9B). pH did not differ between AD eczematous lesions, uninvolved AD skin and healthy skin (Figure 8B).

A negative correlation was found between age and elasticity in all the population (r = −0.383, *p* < 0.001). This correlation was stronger in AD patients (r = −0.494, *p* < 0.001) than in controls (r = −0.266, *p* = 0.092), Figure 10. Moreover, on uninvolved AD skin (Appendix A), SCORAD was positively correlated to pH (r = 0.401, *p* = 0.001) and negatively correlated to SCORAD and TEWL (r = −0.672, *p* < 0.001), SCH (r = −0.504, *p* < 0.001) and elasticity (r = −0.541, *p* < 0.001). A correlation close to significance was found between SCORAD and erythema on uninvolved skin (r = 0.250, *p* = 0.090). No significative correlations were found between SCORAD and temperature (r = 0.002, *p* = 0.968). On AD eczematous lesions (Appendix A), a positive correlation was found between SCORAD and temperature (r = 0.291, *p* = 0.011) and between SCORAD and pH (r = 0.401, *p* < 0.001); and a negative correlation between SCORAD and SCH (r = −0.319, *p* = 0.006) and between SCORAD and elasticity (r = −0.419, *p* = 0.003). A correlation close to significance was found between SCORAD and erythema on eczematous lesions (r = 0.258, *p* = 0.080). No correlation was found between SCORAD and TEWL (r = −0.016, *p* = 0.892). After conducting a linear regression model in AD patients adjusted by age, sex and SCORAD, it was found that elasticity was impaired by a higher age (β = −0.004, *p* < 0.001) and a higher SCORAD (β = −0.003, *p* < 0.001).

## 4. Discussion

This research shows that TEWL, erythema, and pH are increased while SCH and elasticity are reduced in AD patients’ eczematous lesions. Higher SCORAD values are associated with skin barrier dysfunction, reflected in higher pH and temperature and lower SCH and elasticity.

Our data found that TEWL is higher both on AD eczematous lesion and uninvolved AD skin than on healthy skin, like earlier reports [5,27,28]. Previously, it was also found that SCH was decreased on AD lesions [5,29], in agreement with our results. Higher TEWL and lower SCH values in AD patients reflects a skin barrier dysfunction, specifically expressed at lesioned skin [29]. This could be justified by filaggrin mutations or deficiency, as it is the major protein constituent of the stratum corneum and influences keratin filament aggregation [30]. This research also observed that erythema is higher on eczematous lesions than uninvolved AD skin and healthy skin, showing inflammatory changes happening in this disease [31]. Moreover, only in participants ≥ 30 years of age, erythema was higher on uninvolved AD skin than healthy skin in agreement with a higher inflammatory charge observed in elderly AD patients [17].

pH was increased on AD eczematous lesion compared with healthy skin in participants < 30 years of age. It was previously found that AD patients showed higher pH values both on AD eczematous lesions and AD uninvolved skin [32]. Alkalinization of the pH plays a key role in skin barrier dysfunction in AD patients [33]. There are three mechanisms that impact skin acidification: the breakdown of phospholipids to liberate free fatty acids by secretory phospholipase A2, the transport of protons into the extracellular compartment via the action of the sodium–hydrogen exchanger 1 protein, and the generation of free amino acids (AA) via the catabolism of SC structural components [33]. This third mechanism may explain the alkalinization linked to AD, as the main source of AA in the stratum corneum are the filaggrin, filaggrin-2 and Hornerin [34]. These proteins are catabolized to generate free AA, such as trans urocanic acid and pyrrolidone carboxylic acid, AA that make up over half of the constituents of natural moisturizing factor (NMF). NMF plays an important role maintaining skin hydration and skin acidification [35]. So, the deficiency of filaggrin might explain the higher pH in AD patients.

We only found two previous reports that compared elasticity parameters between AD patients and healthy individuals [5,36]. They found a decreased elasticity on AD eczematous lesions than on uninvolved AD skin [5,36]. Collagen and elastin are the main proteins responsible for skin elasticity [37], so the differences in elasticity between AD patients and healthy individuals may reveal that they are other proteins altered in AD, in addition to filaggrin.

Some studies suggested that the immune mechanisms of AD differ between ages [18,38,39]. Young AD patients showed similar or greater epidermal hyperplasia and immune infiltration, decreased filaggrin expression on histology and immunohistochemistry, and activation of the Th2, Th22, and Th1 axes compared with the elderly [38,39]. Young AD patients also had higher induction of Th17-related cytokines, antimicrobials, Th9, IL-31, IL-33, and innate markers than adults [38,39]. In our study, some differences in skin barrier function were observed when stratifying by age. Only in adults ≥ 30 years of age erythema and temperature were higher and elasticity was lower on uninvolved AD skin than healthy skin; while only participants < 30 years of age showed decreased SCH on AD uninvolved skin compared with healthy skin and lower pH on healthy skin compared with uninvolved AD skin and AD eczematous lesions.

Concerning cutaneous homeostasis parameters and AD severity, previously, correlations were observed between TEWL and SCORAD [40], and between skin hydration and SCORAD [41,42]. Furthermore, TEWL values on non-involved AD skin predicted AD development in infants [43]. Cut off points in TEWL and temperature on AD eczematous lesions could be used to assess AD severity [5]. We found that disease severity was positively correlated with temperature and pH and negatively correlated with SCH and elasticity. The NMF is decreased in AD patients with a severe disease [34], likely explaining the correlation between AD severity, pH, and SCH. Higher temperatures reported in more severe AD patients possibly suggest greater vascularization and degree of inflammation [44], which is further supported by several biomarkers such as T helper 2-skewed markers (IL-13, CCL17, CCL22, IL-5), markers of innate activation (IL-18, IL-1α, IL1β, CXCL8), and angiogenesis (Flt-1, vascular endothelial growth factor) [45].

In our study we used the R2 parameter to evaluate elasticity. R2 measures the overall elasticity of the skin, including creep and creep recovery. There are other parameters that also measure elasticity. Previously, positive correlations of R4 and R6, and inverse correlations of R5, R7 and R2 have been associated with age [46], which supports our study’s findings. R7 is the ratio of elastic recovery to the total deformation. It has been observed that R2 parameter represents the gross elasticity of the skin and a higher correlation coefficient of R7 than R2 with age could mean that the elastic fibers degrade more rapidly with aging than do the other components of skin [46].

Our study also revealed that elasticity is impaired by a higher age and a higher SCORAD. Elastin is the major component of elastic fibers, and it is a particularly vulnerable protein because of its slow turnover [47,48]. The normal production of elastic fibers and their integration with proteoglycans, glycosaminoglycans and other extracellular matrix proteins are necessary to preserve a functional skin structure [48]. Elastin production decreases after adolescence and is susceptible to damage from many factors, like environmental exposure or inflammation [48]. Skin aging disrupts the elastic fiber network and decreases collagen, hyaluronic acid, glycosaminoglycans, integrins, and laminin, causing a lack of tissue compliance and structural damage [49]. Moreover, inflammation promotes the recruitment of elastases, elastolytic enzymes that degrade elastin fibers [47]. As AD is an inflammatory skin disease, impairments in the elastic fiber network are expected without intrinsically aged skin [47]. The synergic effect of skin aging and inflammation explains the lower elasticity findings observed in old AD patients.

The results of this study should be considering its limitations: (1) its cross-sectional design; (2) the inclusion of AD patients that were currently being treated that could likely have an impact on skin barrier function. (3) the possibility that AD severity could impact on differences in skin barrier function (though no statistical differences were found and the absolute difference between ages groups was small); (4) the control population was not a random population sample, but rather consisted of volunteer patients being treated for other dermatological conditions; (5) only few children were included in our study despite AD being prevalent in children than adults. This is because the study setting where the research was conducted is mainly dedicated to attending adults’ patients. It would be interesting to conduct further studies to compare differences in skin barrier function between adults and children with AD. The main strength of our study is the high number of participants and the novelty of the objective evaluation of skin barrier function assessing differences between ages.

## 5. Conclusions

In conclusion, this study suggests that skin is impaired by age and AD, reflected mainly in poor elasticity values in older AD patients. Moreover, AD patients ≥ 30 years of age might have higher temperature and erythema and lower elasticity on uninvolved AD skin than healthy volunteers, changes that might not appear in younger participants. Further investigation should be developed to increase the knowledge about the molecular mechanisms related to skin aging and the development of AD.

## Figures and Tables

**Figure 1 life-12-00132-f001:**
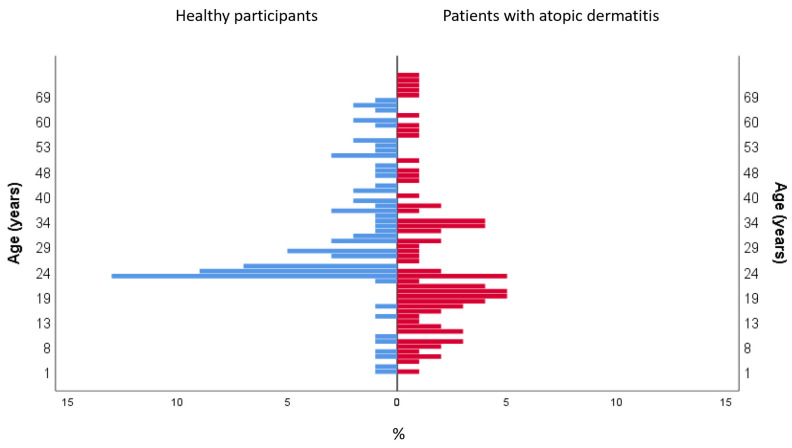
Age distribution in the population.

**Figure 2 life-12-00132-f002:**
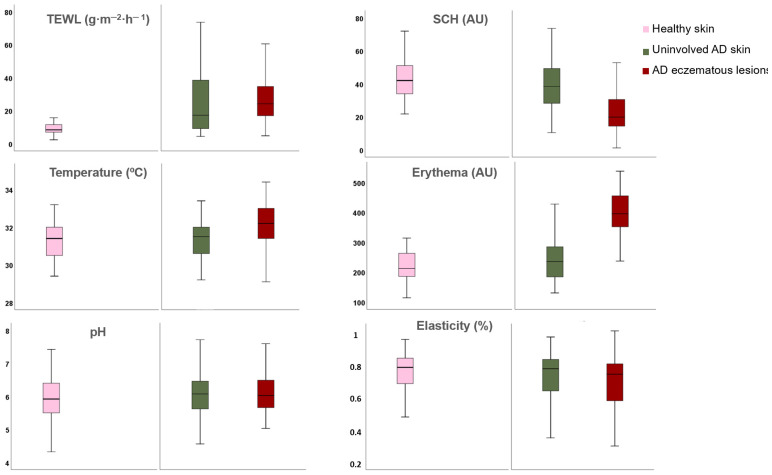
Skin barrier function parameters between patients with atopic dermatitis and healthy volunteers. AD, atopic dermatitis; AU, arbitrary units; SCH, stratum corneum hydration; TEWL, transepidermal water loss.

**Figure 3 life-12-00132-f003:**
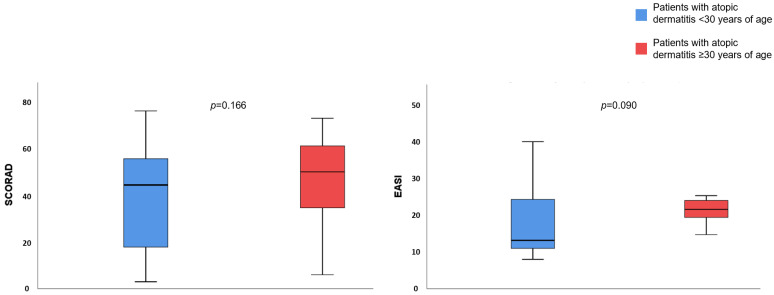
Disease severity depending on the age. *p*-value after using Student’s *t*-test for independent samples to compare disease severity between patients with atopic dermatitis < 30 and ≥ 30 years of age.

**Figure 4 life-12-00132-f004:**
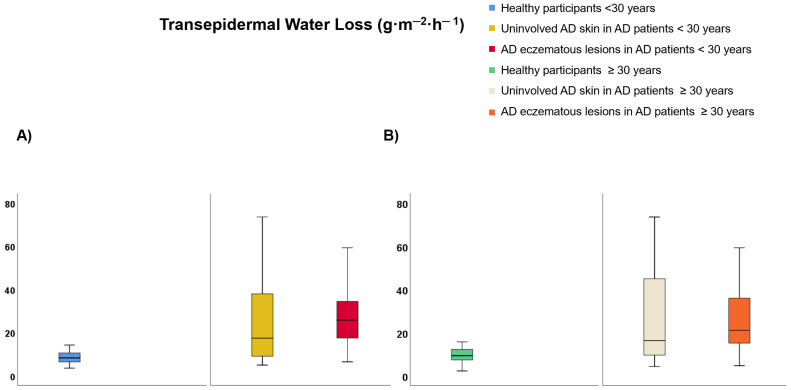
Transepidermal water loss comparing (**A**) patients with atopic dermatitis < 30 and healthy participants < 30, and (**B**) patients with atopic dermatitis ≥ 30 and healthy participants ≥ 30.

**Figure 5 life-12-00132-f005:**
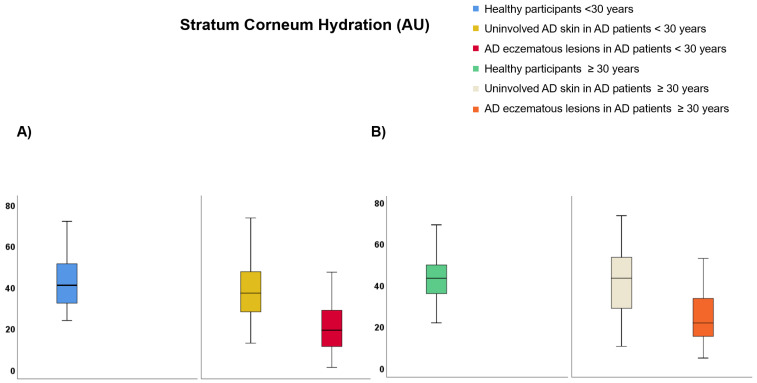
Stratum corneum hydration comparing (**A**) patients with atopic dermatitis < 30 and healthy participants < 30, and (**B**) patients with atopic dermatitis ≥ 30 and healthy participants ≥ 30.

**Figure 6 life-12-00132-f006:**
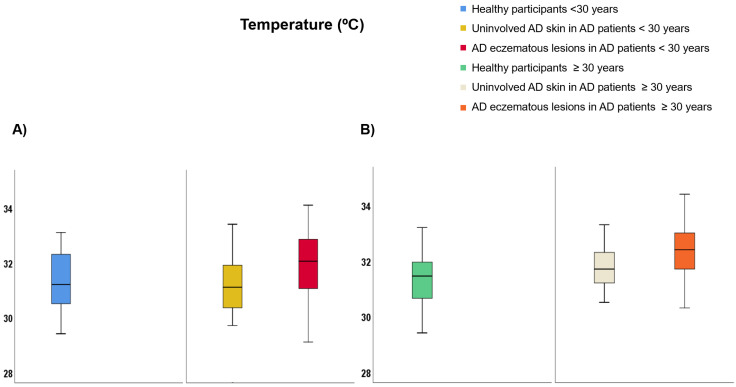
Temperature comparing (**A**) patients with atopic dermatitis < 30 and healthy participants < 30, and (**B**) patients with atopic dermatitis ≥ 30 and healthy participants ≥ 30.

**Figure 7 life-12-00132-f007:**
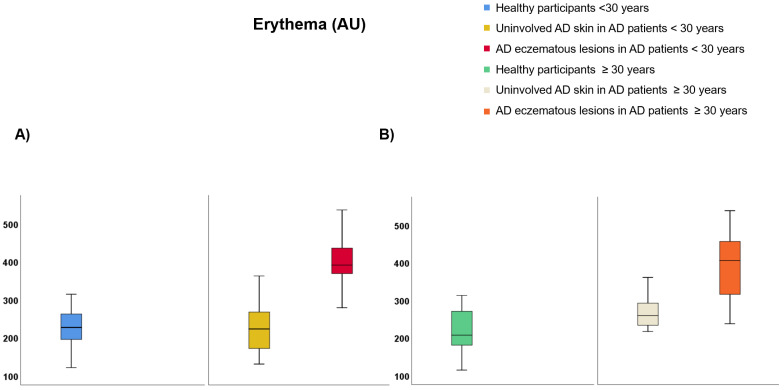
Erythema comparing (**A**) patients with atopic dermatitis < 30 and healthy participants < 30, and (**B**) patients with atopic dermatitis ≥ 30 and healthy participants ≥ 30.

**Figure 8 life-12-00132-f008:**
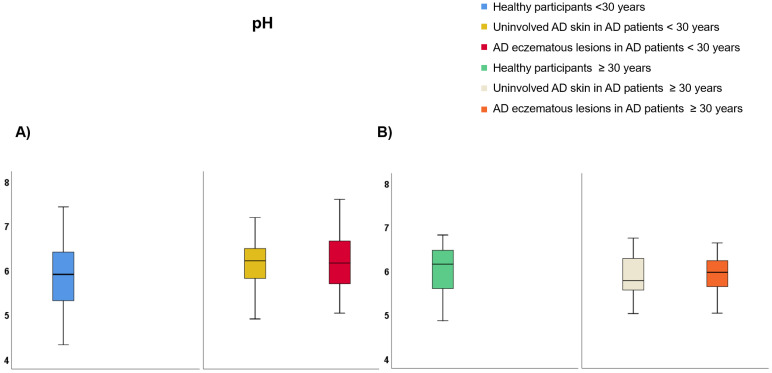
pH comparing (**A**) patients with atopic dermatitis < 30 and healthy participants < 30, and (**B**) patients with atopic dermatitis ≥ 30 and healthy participants ≥ 30.

**Figure 9 life-12-00132-f009:**
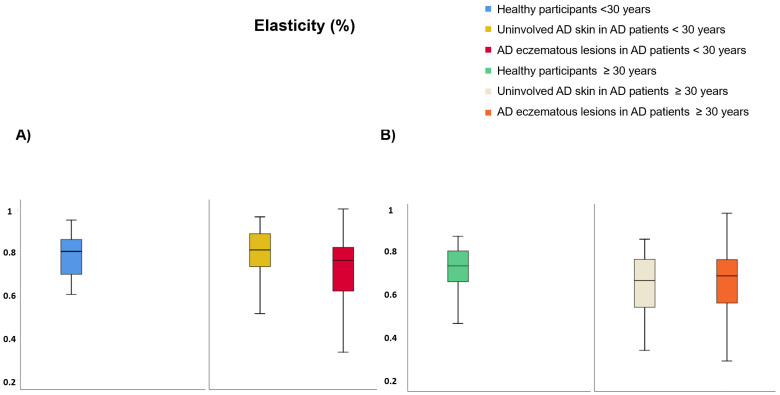
Elasticity comparing (**A**) patients with atopic dermatitis < 30 and healthy participants < 30, and (**B**) patients with atopic dermatitis ≥ 30 and healthy participants ≥ 30.

**Figure 10 life-12-00132-f010:**
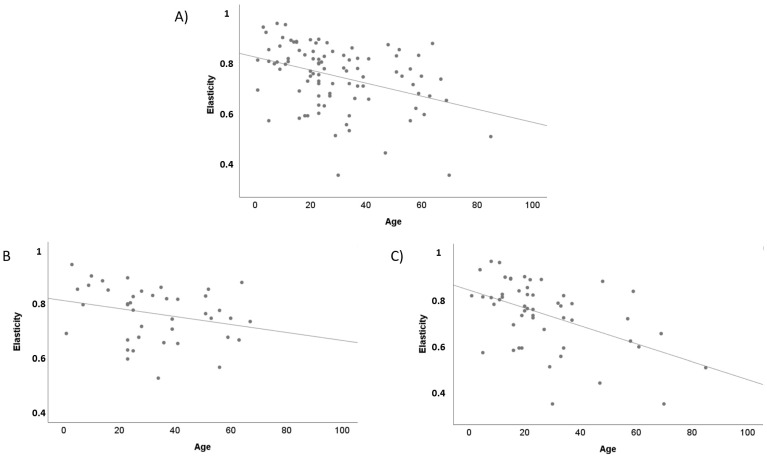
Correlation between age and elasticity. (**A**) In all the population (r = −0.383, *p* < 0.001). (**B**) In healthy volunteers (r = −0.266, *p* = 0.092). (**C**) In atopic dermatitis patients (r = −0.494, *p* < 0.001).

**Table 1 life-12-00132-t001:** Characteristics of the sample.

Sociodemographic Features	Atopic Dermatitis Patients (*n* = 81)	Healthy Participants (*n* = 81)	*p* Value *
**Age (years)**	31.88 (14.92)	27.40 (18.15)	0.088
**Sex (%)**			1.00
- **Female**	53 (65.4%)	53 (65.4%)	
- **Male**	28 (34.6%)	28 (34.6%)	
**Smoking habit (yes)**	14 (17.28%)	6 (7.4%)	0.291
**Alcohol habit (yes)**	26 (32.10%)	24 (29.63%)	0.873
**Emollients use (yes)**	68 (83.95%)	53 (65.43%)	0.080
**Treatment**		-	-
- **Topical treatment**	45 (55.56%)		
- **Systemic treatment**	27 (33.33%)		
- **Biologic drugs**	9 (11.11%)		

Data are expressed as relative (absolute) frequencies and means (standard deviations (SDs)). * *p* value after using Student T test for independent samples to compare continuous variables and the chi-square test to compare categoric data between healthy participants and atopic dermatitis patients.

## Data Availability

The data presented in this study are available on request from the corresponding author.

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
