# Peer review of "Epidermal Barrier Function and Skin Homeostasis in Atopic Dermatitis: The Impact of Age"

_life, 2022, doi:10.3390/life12010132_

Round 1

Reviewer 1 Report

The authors reported the results of a cross-sectional study aiming to investigate skin barrier function changes with age in AD patients.

I found the whole manuscript clear and well written. Moreover,  reported data were interesting due to the poor literature about a comparison of elasticity and other evaluated parameters between AD patients and healthy individuals.

The introduction section provides sufficient background and was well referenced.

The study design and the methods were well resumed in the method section.

Results were well resumed. Tables and figures were clear and related to the results.

In the discussion section, the authors reported an interesting overview of the actual knowledge of probiotic use in acne management, and well-argued the main topic of the study as well as their results, comparing them with already reported data.

Author Response

Thank you very much for all the comments.

Reviewer 2 Report

This was a very interesting and well-orchestrated study. Most of my comments are centered around errors in how data is presented and conclusions that are overstated by the authors. The authors are encouraged to focus on the data and what those data support. Overstating your conclusions as facts creates unnecessary skepticism in an otherwise sound study.  

This journal has a wide audience you must define acronyms before using them (even in the abstract).

The abstract is a little unclear. Line 15-16 established 2 study populations, but the data in lines 18-20 includes three values (xx vs xx vs xx). it is unclear which groups each of those values represents. In line 19 the abstract reaffirms two study groups. You do not establish in the abstract the multiple samples on AD participants. Please clarify which population each of these values represents.

In line 19: I would assume that p<0.014 should read p=0.014

P values do not match between figures, results and abstract sections. Please correct.

What was the age distribution? Were any children included in the study.  The authors noted that age related changes occur in the thirties however the mean study age was in the 30s. The p value (p = 0.08) also suggests that there was some heterogeneity between the AD and healthy participants age populations. Please provide this data in a figure so that the comparability and range of study participants can be assessed.

Figures showing the differences in EASI and SCORAD between age groups should be included. The comparability of EASI and SCORAD scores between the <30 and >30 must be established to differentiate age from disease level.  

Sex, smoking/drinking and skin care habits were evaluated as covariables. This data was not included in this study. How did these factors influence the age-related differences observed? Statements on the influence of each should be included. If these were not evaluated then they should be added to the limitations.

The authors should comment on why there is a difference in SCH between unevolved skin and eczematous lesions, but significant differences in TEWL?

Mean values are reported on temp data for figure 3 but on no other graph. This should be consistent.

Font in some figures is too small to read clearly in some instances.

The data range differs on comparable figures making it difficult to evaluate trends. Please set scales to be the same for each parameter.

Figure 2 and 3 needs to be reorganized. When comparing differences in age between parameters it is better to have figure for the same parameter set beside each other for each age group so that comparisons in the data can be made. With the current organization, each age group is separated requiring readers to flip back and forth between composite figures 2 and 3 to make these comparisons. Reorganize these.  

Use box plots rather than bar graphs as a more illustrative way of showing data trends than bar graphs. Especially when data ranges would not include zero. For example, a bar graph suggests that some temperature values could have been zero but the average was ~31 degrees C. A box more accurately shows the inclusive upper and lower limits of that data.

The authors chose to handpick the data they thought was interesting and did not report on many of the results. Data indicating no significant difference and a low correlation is often as informative as significant differences. Statement on the outcome of these results should be included, even if the figures are in the supplemental material.

Unclear why some correlations were conducted with SCORAD and others with EASI

Lines 244 -245; Omit the 1st two sentences of the discussion.

Line 245-247; References are needed to these statements or if they are in references to this study data these should indicate that data supports these findings.

The following sentences in line 248 should be omitted “This study shows that the whole epidermal barrier is affected in AD patients”. It is inappropriate in a scientific paper to overstate your results or draw conclusions unsupported by the data.

In the discussion and conclusion present tense should be used to interpret results and to discuss the significance and conclusions of the study (“Our data suggest this..). Past tense should be used to summarize the overall findings from the research (“We discovered a new therapeutic target for…”). Future tense should be used to convey perspectives and plans. Please adjust accordingly.

Other age-related trends were reported in the discussion (e.g. line 285) that were not shared in the results. These should be included in the results (especially if they are noteworthy enough to discuss).

The authors noted that the prevalence of AD is higher in children (20% of children relative to 10% in adults; line 58). The results from this study seem to be at odds with that observation. The authors should explain why age-related changes associated with AD could be explained given a higher prevalence in children.

Limitation should include that the control population was not a random population sample, but rather consisted of volunteer patients being treated for other dermatological conditions.

The conclusions should also be rewritten to indicate what the data in this study support, rather than absolutes that cannot be determined given the limitations of this study.

Some copyediting and spacing issues, exist throughout. Extra spaces or missing spaces are the most prevalent issues. Please carefully review the proof copy to insure a high-quality final draft

Link to supplemental data did not work and could not be evaluated.  

Author Response

This was a very interesting and well-orchestrated study. Most of my comments are centered around errors in how data is presented and conclusions that are overstated by the authors. The authors are encouraged to focus on the data and what those data support. Overstating your conclusions as facts creates unnecessary skepticism in an otherwise sound study.  

Thank you for the comments

This journal has a wide audience you must define acronyms before using them (even in the abstract).

We have defined all the acronyms the first time they appear even in the abstract.

The abstract is a little unclear. Line 15-16 established 2 study populations, but the data in lines 18-20 includes three values (xx vs xx vs xx). it is unclear which groups each of those values represents. In line 19 the abstract reaffirms two study groups. You do not establish in the abstract the multiple samples on AD participants. Please clarify which population each of these values represents.

There are two study groups: patients with atopic dermatitis and healthy individuals. Patients with atopic dermatitis were measured at two body sites: eczematous lesions and uninvolved skin. Therefore, we compare three “types of skin”: healthy skin (healthy individuals’ skin), uninvolved AD skin (skin without lesions in AD patients) and AD eczematous lesions (skin with lesions in AD patients). Following your recommendations, we have added this sentence in the abstract: “Healthy volunteers were evaluated on the volar forearm. AD patients were measured on two regions: on an eczematous lesion on the volar forearm and on a non-involved area 5 cm from the affected area”

In line 19: I would assume that p<0.014 should read p=0.014

It is p=0.014. It has been checked in the abstract. Thank you for the comment

P values do not match between figures, results and abstract sections. Please correct.

We have checked the p values between abstract, results and figures. We have also changed the sentences in the abstract regarding skin barrier function to avoid confusion: “TEWL was lower on healthy skin than uninvolved AD skin (9.98 vs 25.51 g·m−2·h−1, p<0.001) and AD eczematous lesions (9.98 vs 28.38 g·m−2·h−1, p<0.001). SCH was lower on AD eczematous lesions than uninvolved AD skin (24.23 vs 39.36 AU, p<0.001) and healthy skin (24.23 vs 44.36 AU, p<0.001). Elasticity was lower on AD eczematous lesions than uninvolved AD skin (0.69 vs 0.74, p= 0.038) and healthy skin (0.69 vs 0.77, p=0.014).” In the abstract, we have only included the correlation and regression model about the relation between skin barrier function and age and have not included information regarding differences in skin barrier function between the two ages group because we did not want to exceed the word limit but if you think it is necessary, we could also include this information. Following your recommendation, we have also checked the figures and the results and have modified the way of expression the differences and p values to avoid confusion. For example, “TEWL was lower on healthy skin than uninvolved AD skin and AD eczematous lesions (9.98 vs 25.51 vs 28.38 g·m−2·h−1, p<0.001) while no differences between uninvolved AD skin and AD eczematous lesion were found” has been changed by “TEWL was lower on healthy skin than uninvolved AD skin (9.98 vs 25.51 g·m−2·h−1, p<0.001) and AD eczematous lesions (9.98 vs 28.38 g·m−2·h−1, p<0.001) while no differences between uninvolved AD skin and AD eczematous lesion were found”. Moreover p-values lower than 0.05 are mentioned in the text but they have been deleted in the new figures to avoid overloading them. All p-values are also provided in the supplementary tables.

What was the age distribution? Were any children included in the study.  The authors noted that age related changes occur in the thirties however the mean study age was in the 30s. The p value (p = 0.08) also suggests that there was some heterogeneity between the AD and healthy participants age populations. Please provide this data in a figure so that the comparability and range of study participants can be assessed.

The mean age of the study population was 29.64 (16.71 SD) years: 31.88 (14.92) years in AD patients and 27.40 (18.15) years in healthy individual. The minimum age was 1 and the maximum age was 85. Children were allowed to participate in the study. We have provided a new figure where the age distribution in each population can be observed: Figure 1. Age distribution in the population.

Figures showing the differences in EASI and SCORAD between age groups should be included. The comparability of EASI and SCORAD scores between the <30 and >30 must be established to differentiate age from disease level.  

Following your recommendation, we have added a new figure and have added information comparing disease severity between age groups. Disease severity was compared between AD patients <30 and ≥30 years, Figure 3. AD patients ≥30 years had a tendentially higher EASI than AD patients <30 years (21.96 vs 17.68, p=0.090) while there were no differences regarding SCORAD (45.57 vs 38.39 respectively, p=0.166). We have also included this issue as a limitation of our study and the following sentence has been added: “The possibility that AD severity could impact on differences in skin barrier function, but no statistical differences were found and the absolute difference between ages groups was small”.

Sex, smoking/drinking and skin care habits were evaluated as covariables. This data was not included in this study. How did these factors influence the age-related differences observed? Statements on the influence of each should be included. If these were not evaluated then they should be added to the limitations.

Sex, smoking/drinking and emollients use were recorded in this study. Comparison between AD patients and healthy individuals regarding these variables is provided in table 1. There were no differences between populations. A tendentially higher emollient use was found in AD patients. If it was a bias of our study, it would be a bias towards the null, meaning that the real differences in skin barrier function between healthy individuals and atopic dermatitis patients are even higher than those observed in our study. We also compared these variables between AD patients <30 and > 30 years of age and did not find any differences. Following you recommendation we have added this information in the results section and have added this sentence: “Sex, smoking/drinking habit, emollients use was similar between the two age groups”. We have also modified this sentence in the material and method section: Sex, age, smoking/alcohol habits and emollient use were collected

Mean values are reported on temp data for figure 3 but on no other graph. This should be consistent.

Mean temperature values have been deleted from this figure

Font in some figures is too small to read clearly in some instances.

We have increased the letter size of the figures. Please let us know if there is still some that is difficult to read.

The data range differs on comparable figures making it difficult to evaluate trends. Please set scales to be the same for each parameter.

All figures comparing the same parameters have been set with the same scale.

Figure 2 and 3 needs to be reorganized. When comparing differences in age between parameters it is better to have figure for the same parameter set beside each other for each age group so that comparisons in the data can be made. With the current organization, each age group is separated requiring readers to flip back and forth between composite figures 2 and 3 to make these comparisons. Reorganize these.  

Figure 2 and 3 have been eliminated. We have constructed a new figure for each parameter and data of different participants <30 and ≥ 30 is provided in the same figure following your recommendations. P values lower than 0.05 are mentioned in the text but they have been deleted in the new figures to avoid overloading them. All p-values are mentioned in the supplementary tables.

Use box plots rather than bar graphs as a more illustrative way of showing data trends than bar graphs. Especially when data ranges would not include zero. For example, a bar graph suggests that some temperature values could have been zero but the average was ~31 degrees C. A box more accurately shows the inclusive upper and lower limits of that data.

We have changed all the figures previously provided as bar graphs to box plots as recommended.

The authors chose to handpick the data they thought was interesting and did not report on many of the results. Data indicating no significant difference and a low correlation is often as informative as significant differences. Statement on the outcome of these results should be included, even if the figures are in the supplemental material.

We have included information regarding differences between all skin barrier function parameters. We have selected SCORAD as the severity scale because there were no differences regarding this in our study population. Moreover, SOCRAD also evaluate subjective symptoms such as pruritus that could also impact on skin barrier function. The following sentences have been added: “Moreover, on uninvolved AD skin (Figure 11), SCORAD was positively correlated to pH (r=0.401, p=0.001) and negatively correlated to SCORAD and TEWL (r=-0.672, p<0.001), SCH (r=-0.504, p<0.001) and elasticity (r=-0.541, p<0.001). A correlation close to significance was found between SCORAD and erythema on uninvolved skin (r=0.250, p=0.090). No significative correlations were found between SCORAD and temperature (r=0.002, p=0.968). On AD eczematous lesion (Figure 12), a positive correlation was found between SCORAD and temperature (r=0.291, p=0.011) and between SCORAD and pH (r=0.401, p<0.001); and a negative correlation between SCORAD and SCH (r=-0.319, p=0.006) and between SCORAD and elasticity (r=-0.419, p=0.003). A correlation close to significance was found between SCORAD and erythema on eczematous lesions (r=0.258, p=0.080). No correlation was found between SCORAD and TEWL (r=-0.016, p=0.892)”.

Unclear why some correlations were conducted with SCORAD and others with EASI

We have selected SCORAD as the severity scale because there were no differences regarding this in our study population. Moreover, SOCRAD also evaluate subjective symptoms such as pruritus that could also impact on skin barrier function.

Lines 244 -245; Omit the 1st two sentences of the discussion.

They have been omitted.

Line 245-247; References are needed to these statements or if they are in references to this study data these should indicate that data supports these findings.

This sentence refers to this study data. We have added the following words: “This research shows that…”

The following sentences in line 248 should be omitted “This study shows that the whole epidermal barrier is affected in AD patients”. It is inappropriate in a scientific paper to overstate your results or draw conclusions unsupported by the data.

We have deleted this sentence.

In the discussion and conclusion present tense should be used to interpret results and to discuss the significance and conclusions of the study (“Our data suggest this..). Past tense should be used to summarize the overall findings from the research (“We discovered a new therapeutic target for…”). Future tense should be used to convey perspectives and plans. Please adjust accordingly.

We have changed the tense following your recommendations. Please if there are more sentences that we should change, let us know.

Other age-related trends were reported in the discussion (e.g. line 285) that were not shared in the results. These should be included in the results (especially if they are noteworthy enough to discuss).

We have checked this sentence: Only in adults ≥ 30 years of age, erythema and temperature were higher and elasticity was lower on uninvolved AD skin than healthy skin; while only participants<30 years of age showed decreased SCH on AD uninvolved skin compared to healthy skin and lower pH on healthy skin compared to uninvolved AD skin and AD eczematous le-sions. We have also underlined where they appear in the results section: Regarding participants <30 years of age (Table S2), TEWL was lower on healthy skin than uninvolved AD skin (9.99 vs 25.69 g·m−2·h−1, p<0.001) and AD eczematous lesions (9.99 vs 28.32 g·m−2·h−1, p<0.001) but it did not differ between AD eczematous lesions and uninvolved AD skin (Figure 4.A). SCH was lower on AD eczematous le-sions than uninvolved AD skin (22.47 vs 38.22 AU, p<0.001) and healthy individuals (22.47 vs 48.13 AU, p<0.001) and it was also lower on uninvolved AD skin than healthy individuals (38.22 vs 48.13 AU, p=0.030) (Figure 5.A). Temperature was higher on AD eczematous lesions than uninvolved AD skin (31.94 vs 31.05, p<0.001) and healthy skin (31.94 vs 31.07ºC, p=0.010) but it did not differ between uninvolved AD skin and healthy skin (Figure 6.A). Erythema was higher on AD eczematous lesions than unin-volved AD skin (388.80 vs 226.75 AU, p<0.001) and healthy skin (388.80 vs 226.73 AU, p<0.001), but it did not differ between uninvolved AD skin and healthy skin (Figure 7.A). pH was lower on healthy volunteers than uninvolved AD skin (5.81 vs 6.16, p=0.029) and AD eczematous lesions (5.81 vs 6.23, p=0.008) but it did not differ be-tween uninvolved AD skin and AD eczematous lesions (Figure 8.A). Elasticity was lower on AD eczematous lesions than uninvolved AD skin (0.71 vs 0.78, p=0.003) and on healthy skin (0.71 vs 0.79, p=0.040) but it did not differ between uninvolved AD skin and healthy skin (Figure 9.A). Concerning adults ≥ 30 years of age (Table S3), TEWL was lower on healthy skin than uninvolved AD skin (10.75 vs 25.18 g·m−2·h−1, p=0.001) and AD eczematous lesions (10.75 vs 28.48 g·m−2·h−1, p<0.001) but it did not differ between uninvolved AD skin and AD eczematous lesions (Figure 4.B). SCH was lower on AD eczematous lesions than on uninvolved AD skin (27.63 vs 41.56 AU, p=0.001) and healthy skin (27.63 vs 43.16 AU, p<0.001) but it did not differ between uninvolved AD skin and healthy skin (Figure 5.B). Temperature was higher on AD eczematous lesions than uninvolved AD skin (32.30 vs 31.70ºC, p=0.011) and healthy skin (32.30 vs 31.17ºC, p<0.001) and it was also higher on uninvolved AD skin than healthy skin (31.70 vs 31.17ºC, p=0.042) (Figure 6.B). Erythema was higher on AD eczematous lesions than uninvolved AD skin (395.32 vs 263.82 AU, p<0.001) and healthy skin (395.32 vs 214.08 AU, p<0.001) and it was also higher on uninvolved AD skin than healthy skin (263.82 vs 214.08 AU, p=0.004) (Figure 7.B). Elasticity was higher on healthy skin than uninvolved AD skin (0.74 vs 0.65, p=0.043) and AD eczematous lesions (0.74 vs 0.65, p=0.040) but it did not differ be-tween AD eczematous lesions and uninvolved AD skin (Figure 9.B). pH did not differ between AD eczematous lesions, uninvolved AD skin and healthy skin (Figure 8.B).

The authors noted that the prevalence of AD is higher in children (20% of children relative to 10% in adults; line 58). The results from this study seem to be at odds with that observation. The authors should explain why age-related changes associated with AD could be explained given a higher prevalence in children.

We collect patients from our Dermatology Department where most of the patients attended are adults. We attend children in another area where we do not have the tools needed to assess these measures. It would be interesting to include more children and more participants to evaluate differences in skin barrier function between different age range. Following your recommendations, we have added this aspect as one limitation of our study: “only few children were included in our study despite AD is more prevalent in children than adults. This is explained because the study setting where the research was conducted is mainly dedicated to attend adults’ patients. It would be interesting to conduct further studies to compare differences in skin barrier function between adults and children with AD”.

Limitation should include that the control population was not a random population sample, but rather consisted of volunteer patients being treated for other dermatological conditions.

We have added this as a limitation of our study.

The conclusions should also be rewritten to indicate what the data in this study support, rather than absolutes that cannot be determined given the limitations of this study.

We have modified the conclusion following your recommendations: “In conclusion, this study suggests that skin is impaired by age and AD, reflected mainly in poor elasticity values in older AD patients. Moreover, AD patients≥30 years of age might have higher temperature and erythema and lower elasticity on unin-volved AD skin than healthy volunteers, changes that might not appear in younger participants. Further investigation should be developed to increase the knowledge about the molecular mechanism related to skin aging and the development of AD.”

Some copyediting and spacing issues, exist throughout. Extra spaces or missing spaces are the most prevalent issues. Please carefully review the proof copy to insure a high-quality final draft

We have carefully review the proof copy to avoid this mistake

Link to supplemental data did not work and could not be evaluated.

We have submitted it again and added as supplementary files. Moreover, we have added them at the end of the manuscript to avoid any inconvenience.  
